# Influence of Frequent Freeze–Thaw Cycles on Performance of Asphalt Pavement in High-Cold and High-Altitude Areas

**Haibo Cao** [1,2], **Tuanjie Chen** [2,3,*], **Hongzhou Zhu** [1] **and Haisheng Ren** [3]

1   School of Civil Engineering, Chongqing Jiaotong University, Chongqing 400741, China; bohaichb@163.com (H.C.); zhuhongzhouchina@cqjtu.edu.cn (H.Z.)
2   CCCC First Highway Consultants Co., Ltd., Xi'an 710075, China
3   Intelligent Transport System Research Center, Southeast University, Nanjing 211189, China; ren_hs510@seu.edu.cn
*   Correspondence: chentuanj@163.com

**Abstract:** This study explores the temperature changes and freeze–thaw cycles in certain typical high-altitude areas, finding that these areas encounter more than 120, or even more than 200, freeze–thaw cycles per year. Such frequent freeze–thaw cycles deliver significant impact on the performance of asphalt pavements, with cracks becoming a typical problem in high-altitude areas. Such factors as cold weather, large temperature differences, and frequent freeze–thaw cycles have adverse effects on the stress of asphalt pavement materials, resulting in cracks in pavements. By simulating the conditions of such frequent freeze–thaw cycles, this study explores the law of changes in the performance of roads made from asphalt and asphalt mixtures, as well as the low-temperature crack resistance properties of asphalt and asphalt mixtures in frequent freeze–thaw cycles. It is found that the performance of the three different types of asphalt binders used in the test shows basically no change after 50 freeze–thaw cycles, and the asphalt types have a significant effect on the low-temperature performance of asphalt mixtures. The modified asphalt shows a higher viscosity than the matrix asphalt, with better toughness than that of the matrix asphalt at low temperature. Frequent freeze–thaw cycles significantly influence the low-temperature splitting tensile strength and water stability of asphalt mixtures; with increased freeze–thaw cycles, the splitting strength and freeze–thaw splitting tensile strength ratio will gradually decrease to a significant level. The freeze–thaw conditions are found delivering remarkable influence on the low-temperature splitting tensile strength and water stability of asphalt mixtures. The research results of this study provide a basis for the selection of asphalt pavement materials as well as the optimal design of mixtures in high-altitude area like the Qinghai-Tibet Plateau.

**Keywords:** high altitude area; low temperature performance; frequent freeze–thaw cycles; crack resistance





## 1. Introduction

With an average altitude of more than 4000 m, the Qinghai-Tibet Plateau is a typical high-cold and high-altitude area, showcasing such significant plateau climate characteristics as long low-temperature durations, high freeze–thaw cycle frequencies, and strong solar radiation. As a result, asphalt layers in this area are subject to temperature fatigue cracking and extreme temperature freezing, leading to such problems as cracking and asphalt layer cracking due to factors including semi-rigid bases and permafrost subgrades. These problems are widespread and difficult to eradicate; especially, frequent and violent freeze–thaw cycles can greatly accelerate the degradation of pavement performance. With increased service life of the roads, the cost of road maintenance will also go up, thus restricting the development of highways in high-cold and high-altitude areas, including the Qinghai-Tibet Plateau. A survey conducted on pavement usage on the G109 Qinghai-Tibet National Highway found that the typical problems along the highway included transverse

cracks, longitudinal cracks, network cracks, potholes, and so on. The cracked asphalt pavements will allow surface water to penetrate into the cracks. Coupled with the action of low temperatures and frequent freezing and thawing, water will be frozen and accumulated in the mixtures, leading to frost heaving and looseness. Continuous development will soften the base layer and cause water damage of the surface layer. Under repeated actions on traffic loads, there will be further serious secondary pavement problems, such as frost boiling, potholes, net cracks, and so on.

The factors affecting the low-temperature cracking resistance of asphalt mixtures can be divided into two types: internal and external factors. The former showcases the material characteristics and structural characteristics of asphalt mixtures, including aggregate type, asphalt type, asphalt content, gradation type, external admixture materials, etc. The latter factors include traffic loads, environmental factors, etc. The low-temperature cracking resistance of asphalt mixtures depends to a large extent on the low-temperature properties of asphalt materials, the bonding strength of asphalt and mineral aggregates, the type of gradation, and the uniformity of asphalt mixtures. In a strategic highway research program conducted in the United States, the restrained sample temperature stress test and the indirect tensile test were made to evaluate the low-temperature crack resistance of asphalt concrete. The indirect tensile test (IDT) is currently most widely used abroad to characterize the low-temperature performance of asphalt mixtures. The cracking test method is necessary in mechanical empirical pavement design and widely used by Strategic Highway Research Program (SHRP), American Association of State Highway and Transportation Officials (AASHTO), and National Cooperative Highway Research Program (NCHRP) to predict the low-temperature tensile strength and creep compliance of mixtures. Other evaluation indicators can also be obtained through an IDT test. For example, fracture energy and fracture work are used as evaluation indicators to evaluate fatigue cracking of asphalt mixtures [1–3]. Hao et al. used the low-temperature crack resistance coefficient to evaluate the crack resistance of asphalt mixtures, finding that the higher the crack resistance coefficient, the better the low-temperature crack resistance [4]. The effects of different aging degrees on the low-temperature properties of asphalt mixtures have also been studied, with the low-temperature crack resistance properties of asphalt mixtures under different aging conditions identified [5,6]. Francesca Russo et al. [7] focused on the investigation of the rheological properties using a dynamic shear rheometer and carrying out a frequency sweep test at temperatures ranging from 0 to 50 °C in increments of 10 °C.

In a related study on the generation process and influencing factors of cracks when asphalt pavements are subjected to temperature fatigue, Mahboub et al. demonstrated that the development process of cracks has a great impact on the fatigue life of pavements [8] by analyzing the calculation method of the loss energy in the temperature fatigue process of asphalt mixtures under different load fatigue forms. Analyses of fatigue mechanisms of road surfaces showcase that with increased temperature fatigue actions, the damage degree will also increase, and the temperature fatigue can be regarded as a thermal fatigue and low-temperature cracking. Based on these results, the corresponding temperature fatigue life equation was established; the temperature fatigue life of asphalt concrete was examined in low-frequency loading frequency temperature fatigue tests; and the temperature fatigue damage model based on dissipated energy was used to analyze the test results. The damage expressed by the sub-meter dissipative energy has a good linear relationship with the plastic strain of asphalt mixtures [9–14]. Using a numerical simulation finite element model based on fracture mechanics, Fu et al. analyzed the propagation process of road surface cracks and base reflective cracks, finding that both the nonlinear relationship of stress intensity factor and the crack propagation speed would mount with the decrease of the reference temperature [15]. Zhan et al. applied thermal-mechanical coupling solution technology to address the temperature stress surface crack problem of asphalt pavements under the action of low temperatures and large temperature differences. Numerical analyses demonstrate that large temperature differences are an important cause of asphalt pavement damage in high-cold areas [16].

Given the special climatic characteristics, such as high cold and large temperature differences, in high-cold and high-altitude areas, the indoor flexural tensile strength test on asphalt mixtures has been carried out to study the law of changes in flexural tensile strength of asphalt mixtures affected by gradation, oil-stone ratio, temperature, and other factors. The test showed that the effects of the materials and the types of the mixtures have an impact on the performance of the mixtures [17,18]. According to the influence of changes in the composition factors of asphalt mixture materials on their high- and low-temperature performance, water stability, and other elements in the areas with large temperature differences, the order of the sensitivity of the factors affecting such performance of the mixtures was determined [19,20]. The effects of asphalt types and gradation types on the water stability and high-temperature performance of asphalt mixtures in the Qinghai-Tibet high-cold areas were explored with the Marshall test, freeze–thaw splitting test, and rutting test [21]. In addition, Tang et al. verified the linear viscoelastic mechanical behavior of asphalt binders by simulating the creep and rheological properties of asphalt binders before and after aging under extreme temperature conditions [22]. Combined with the characteristics of the Tibet area, it is proposed that high-altitude and low-temperature climates are the key factor affecting pavement structures, whereas the adaptability of typical pavement structures in high-cold and high-altitude areas is examined in mineral gradation ranges [23–25].

It can be concluded from the above research that asphalt mixtures will bear temperature damage when the temperature cycle changes; such damage will have a greater impact at low reference temperatures. Therefore, the impact of asphalt mixtures cannot be ignored. At present, the research on the crack resistance of asphalt mixtures in high-cold and high-altitude areas generally adopts a low-temperature bending test, which has a low correlation with regional climate characteristics and a relatively single evaluation index. Few studies have been done on the crack resistance of materials. Based on the climatic and environmental characteristics of high-cold and high-altitude areas as well as cracks and problems in asphalt pavements, this study will, by simulating cold conditions, large temperature differences, frequent freeze–thaw cycles, changing asphalt types, grading, and other factors, implement some asphalt performance tests, thermal stress restrained specimen tests (TSRST), and freeze–thaw splitting tests, so as to explore the change law of road performance of asphalt and asphalt mixtures and examine the low-temperature crack resistance performance of asphalt mixtures under the condition of frequent freeze–thaw cycles. The research results can provide a basis for selecting asphalt pavement surface materials and get the optimal design of mixtures in high-cold and high-altitude areas.

## 2. Freeze–Thaw Cycles and Pavement Cracking in High-Altitude Areas

### 2.1. Statistics of Freeze–Thaw Cycle in High-Altitude Areas

The average altitude of the Qinghai-Tibet Plateau is higher than 4000 m, with low annual average temperatures and large temperature differences between day and night. As shown in Figure 1, for the temperature changes on a typical sunny day in Nagqu in January 2021, the lowest temperature appeared at 6:00 to 7:00 in the morning, whereas the highest temperature appeared at approximately 14:00 in the afternoon. Taking the daily maximum temperature higher than 0 °C and the daily minimum temperature lower than 0 °C as the freeze–thaw conditions for pavements, respectively, it can be found that one freeze–thaw cycle occurs per day.

In order to identify the characteristics of freeze–thaw cycles in high-cold and high-altitude areas, the observation data in three temperature observation stations in Wudaoliang (altitude 4680 m), Nagqu (altitude 4350 m), and Lhasa (altitude 3650 m) are used. The period from the daily maximum temperature higher than 0 °C to the minimum temperature lower than 0 °C is taken as one freeze–thaw cycle; and the number of freeze–thaw cycles in the three observation stations spanning 36 years from 1980 to 2015 is counted, with the statistical results shown in Figure 2, which shows that the annual freeze–thaw cycles in this high-cold and high-altitude area mostly exceed 120. In addition, with the increase in

altitude, the number of freeze–thaw cycles would show an increasing trend. The melt cycle will inevitably impose an adverse effect on the asphalt pavements in the area.

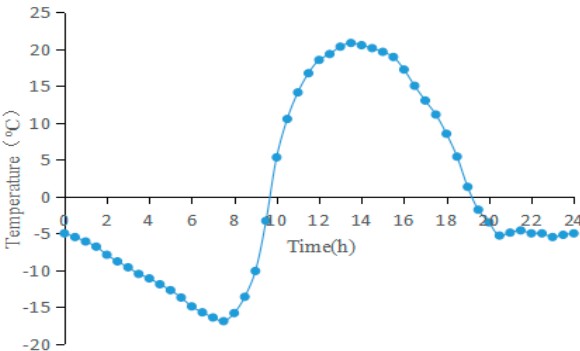

**Figure 1.** The trend of daily temperature variation in Nagqu, Tibet in January.

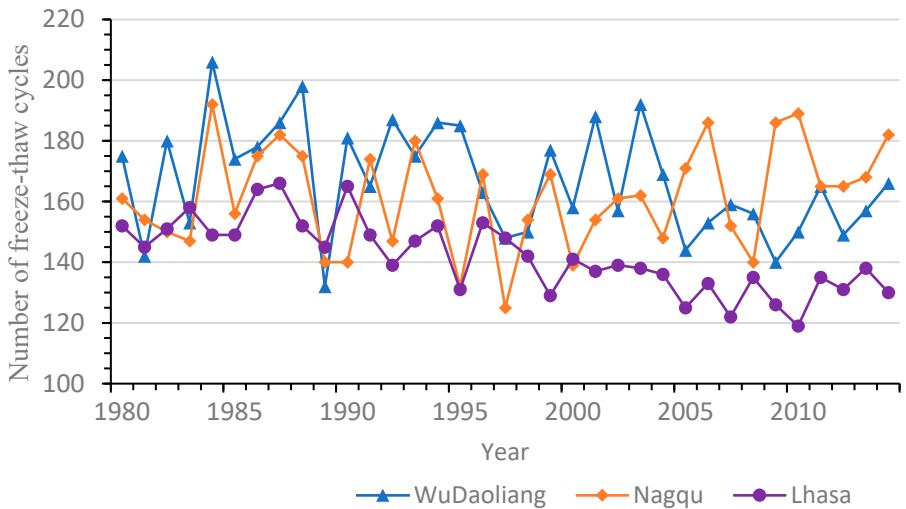

**Figure 2.** Number of freeze–thaw cycles in typical high-cold and high-altitude areas.

### 2.2. Investigation of Pavement Cracking in High Altitude Areas

The road conditions of Gongyu Expressway, G214, G109, and Lhasa-Gongga Airport Special Expressway in high-cold and high-altitude areas were investigated, and it was found that the main problem types of asphalt pavements include cracks, subsidence, ruts, potholes, etc., among which cracks are the most prominent problems. The main performance is shown in Figure 3.

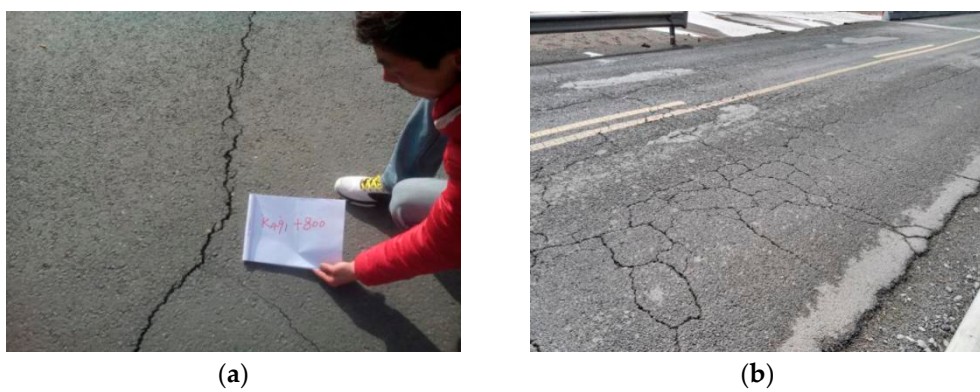

**Figure 3.** Lateral cracking and network cracking problems of asphalt pavement. (**a**) is transverse crack; (**b**) is net crack.

The reasons for the cracks are analyzed. Because the project is located in a high-cold and high-altitude zone, there is a big temperature difference between day and night. Under the conditions of cold, large temperature differences, and frequent freeze–thaw cycles, when the ambient temperature decreases, the asphalt pavement will become warmer inside. When the temperature-shrinkage stress inside the pavement exceeds the ultimate tensile strength of asphalt concrete, the pavement will become cracked. In addition, the cracking of the asphalt surface due to the cracking of the base layer is also a cause of such cracks.

## 3. Experimental Programs

### 3.1. Materials

Based on the types of asphalt used in the high-cold and high-altitude areas in China, No. 90 matrix asphalt, No. 110 matrix asphalt, and SBS modified asphalt are adopted for comparative analysis in the study. The specific technical indicators of such asphalt are tested and summarized in Table 1.

**Table 1.** Technical index value of asphalt.

| Test Content | | Unit | Asphalt Type | | |
|---|---|---|---|---|---|
| | | | No. 90 | No. 110 | SBS Modified |
| Penetration (25 °C) | | 0.1 mm | 89.3 | 100.8 | 73.3 |
| Softening point (Ring-ball method) | | °C | 46.0 | 43.4 | 64.0 |
| Ductility (10 °C) | | cm | 92.0 | 125.0 | 79.0 |
| Density (25 °C)/ | | g·cm$^{-3}$ | 0.998 | 0.977 | 1.101 |
| Solubility (trichloroethylene) | | % | 99.8 | 99.87 | 99.21 |
| RTFOT (163 °C) | Quality loss | % | 0.23 | 0.12 | 0.06 |
| | Penetration ratio | % | 77 | 74 | 73 |
| | Ductility (10 °C) | cm | 32 | 25 | 26 |
| | Softening point | °C | 50.3 | 52.3 | 68.7 |

The coarse aggregate is basalt crushed stone, with its main technical indicators shown in Table 2, which meet the technical requirements of "Technical Specification for Highway Asphalt Pavement Construction" (JTG F40-2004) [26].

**Table 2.** Technical index value of coarse aggregate.

| Test Content | Technical Requirement | Aggregate Specifications/mm | | | |
|---|---|---|---|---|---|
| | | 13.2–16 | 9.5–13.2 | 4.75–9.5 | 2.36–4.75 |
| Apparent relative density | ≥2.6 | 2.954 | 2.956 | 2.948 | 2.932 |
| Needle-like content/% | ≤15 | 5 | 6 | 3 | / |
| Soft stone content/% | ≤3 | 1.8 | 0.8 | / | / |
| Crushing value of stone/% | ≤26 | 14 | 13 | / | / |
| Sturdiness/% | ≤12 | 3 | 3 | 3 | 3 |
| Water absorption/% | ≤2 | 0.82 | 0.79 | 0.91 | 0.76 |

The fine aggregate is made of 0~3 mm machine-made sand, and the filler is limestone ore powder. All the technical indicators of the fine aggregate and the ore powder meet the requirements of "Technical Specification for Highway Asphalt Pavement Construction" (JTG F40-2004).

### 3.2. Mix Design

Two types of asphalt mixtures, i.e., AC-13 and AC-16, are selected in this study, with the design gradation shown in Table 3.

**Table 3.** Design gradation of asphalt mixture.

| Mix Type | Mass Percentage (%) Passing through Each Hole Sieve (mm) | | | | | | | | |
|---|---|---|---|---|---|---|---|---|---|
| | **16.0** | **13.2** | **9.5** | **4.75** | **2.36** | **1.18** | **0.30** | **0.15** | **0.075** |
| AC-13 | 100 | 94.2 | 76.4 | 44.2 | 31 | 24 | 12.3 | 9.5 | 7.4 |
| AC-16 | 96.9 | 87.6 | 73.7 | 52.9 | 29.2 | 19.7 | 10.1 | 8.3 | 7.2 |

No. 90 matrix asphalt and SBS modified asphalt are used to prepare asphalt mixture specimens, and the oil-to-stone ratios of matrix asphalt AC-13 and SBS-modified AC-13 are 5.1% and 5.2%, respectively. The best oil-to-stone ratio of SBS-modified AC-16 is 5.1%.

### 3.3. Test Methods

#### 3.3.1. Freeze–Thaw Cycles Method

According to the temperature distribution (−35 °C to 50 °C) in the high-cold and high-altitude areas, the temperature condition of the freeze–thaw cycle test is set as −35 °C to 50 °C. Different asphalt samples and asphalt mixture samples are placed in an environmental test chamber. When the temperature in the chamber reaches 50 °C, we kept the temperature constant for 1.5 h. We then set the cooling rate (8.0 °C/h) to reduce the temperature in the environmental test chamber, waited for it to drop to −35 °C, and kept it constant for 1.5 h. Next, we set the heating rate (8.0 °C/h) to raise the temperature in the environmental test chamber; when it reaches 50 °C, it was kept constant for 1.5 h. The above freeze–thaw cycle test steps was repeated for 10, 20, 30, and 50 times, separately.

#### 3.3.2. Asphalt Performance Test

According to the "Standard Test Methods of Bitumen and Bituminous Mixtures for Highway Engineering" (JTG E20-2011) [27], tests for the asphalt softening point, penetration, and ductility indexes are performed on the original asphalt and the asphalt samples after freeze–thaw cycles.

The BBR test uses the bending beam rheometer device, and the test is a method to better evaluate the stiffness and creep rate of asphalt at low temperatures, as shown in Figure 4. This test applies the theory of engineering beams to measure the stiffness of the asphalt trabecular specimen under creep load and simulates the creep load with the stress generated in the road surface when the temperature drops; two evaluation parameters, i.e., creep stiffness and m value, are obtained through experiments. Creep stiffness indicates the ability of asphalt to resist permanent deformation, whereas the m value indicates the rate of change of asphalt stiffness under load. The asphalt trabeculae are formed in a rectangular aluminum mold with a size of 125 mm × 12.5 mm × 6.25 mm. Before the test, the asphalt trabeculae are placed in a constant temperature bath for 60 min. The samples are then subjected to BBR test at a temperature of −15 °C.

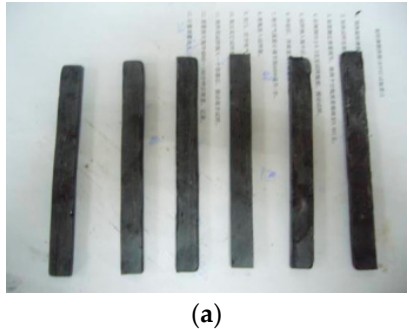 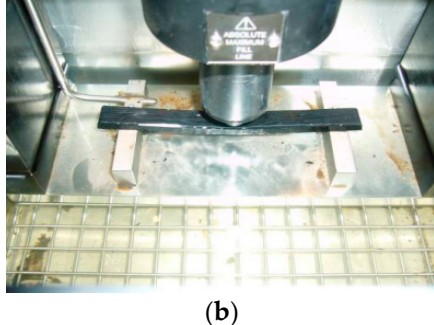

| (**a**) | (**b**) |
|---|---|

**Figure 4.** Low temperature BBR test. (**a**) is BBR test sample; (**b**) is BBR test loading.

The DSR test uses the dynamic shear rheometer device, and the test is implemented to test the viscosity and elasticity of asphalt mortars by measuring the viscous and elastic

properties of thin asphalt cement samples sandwiched between a shock plate and a fixed plate. The complex modulus G* and phase angle δ of asphalt samples are measured with a CSA dynamic shear rheometer, as shown in Figure 5, and the rutting factor G*/sinδ is calculated. The DSR test is carried out on the original asphalt of No. 110 and the asphalt samples after freeze–thaw cycles. This test is mainly purposed to measure the temperature sweep. The temperatures used in the temperature sweep test are 46, 52, 58, and 64 °C; the load frequency is $\omega$ = 10 rad/s; and the strain control mode is used during the test, with $\gamma$ = 12%.

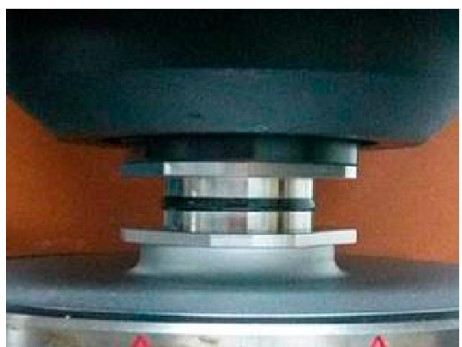

**Figure 5.** DSR test.

### 3.3.3. Thermal Stress Restrained Specimen Test (TSRST)

The TSRST test can truly simulate the actual temperature changes and the actual stress condition of mixtures, so it can truly reflect the low-temperature cracking performance of asphalt mixtures. The test equipment and the fracture states of the samples are shown in Figure 6. Before the test, the trabecular specimen is suspended inside the restraint instrument; four temperature sensors are attached to the four surfaces of the specimen, so as to observe its temperatures in real time; and one temperature sensor is hung on the fixing clip, so as to measure the environment in the restraint instrument. Temperature displacement sensors are installed on both sides of the specimen, which is pulled tight by manual pulleys. During the test, a liquid nitrogen tank continuously transports liquid nitrogen into the restraint instrument. As the temperature decreases, the trabecular specimen will continue to shrink. When the temperature-shrinkage stress of the specimen gets greater than the ultimate tensile strength of the trabecular, the specimen will become cracked.

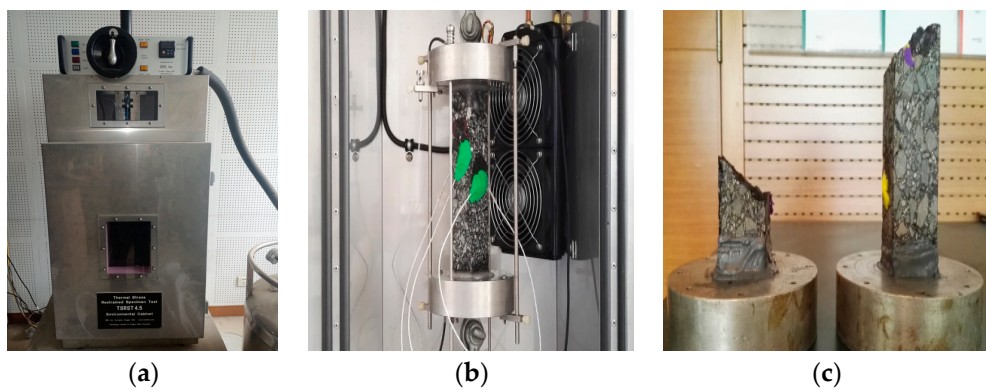

(**a**)　　　　　　　　　　　(**b**)　　　　　　　　　　　(**c**)

**Figure 6.** TSRST sample and fracture state. (**a**) is TSRST equipment; (**b**) is TSRST loading; (**c**) is TSRST sample.

Three kinds of asphalt mixtures are adopted, namely matrix asphalt AC-13, SBS modified asphalt AC-13, and SBS modified asphalt AC-16, which are prepared as 40 mm ×40 mm × 200 mm trabecular specimens. During the cutting process, efforts are made to keep the four edges and corners of the specimen vertical. In order to firmly attach the upper and lower planes

of the test sample to the chassis of the TSRST experimental instrument, Devcon plastic steel repair agent is utilize to bond the two ends of the test sample to the chassis of the instrument. A self-pressurized liquid nitrogen tank is adopted for cooling at a rate of 10 °C/h. The lowest temperature in the test can reach −50 °C. During the cooling process, the test will stop when the specimen breaks down in the middle of the sample. If the specimen does not break, the environment temperature is kept at −50 °C for one hour, and then the test will automatically stop. The freeze–break temperature and freeze–break stress of the asphalt mixture are tested.

3.3.4. Freeze–Thaw Splitting Test

As one of the most widely used test methods, the splitting test characterizes the low-temperature cracking of asphalt mixtures. The freeze–thaw splitting test is used to perform freeze–thaw cycles on asphalt mixtures under specified conditions, so as to determine their strength ratio before and after water damage. The test is carried out by referencing the Standard Test Methods of Bitumen and Bituminous Mixtures for Highway Engineering (JTG E20-2011).

Taking three kinds of asphalt mixtures, namely matrix asphalt AC-13, SBS modified asphalt AC-13, and SBS modified asphalt AC-16, as the research objects, the Marshall test compactor is used to prepare the test samples. The diameter of the test sample is 101.6 mm, and the height is 63.5 mm. The test samples are first vacuum-saturated and then packed into plastic bags; multiple freeze–thaw cycles are performed according to the set conditions. The split tensile strength and freeze–thaw split tensile strength ratio of the asphalt mixtures are tested at a test temperature of −15 °C. The test process is illustrated in Figure 7.

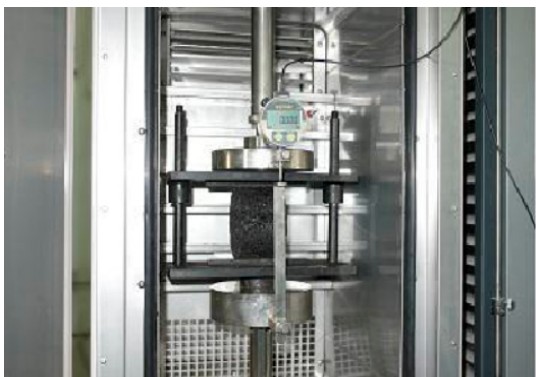

**Figure 7.** Low temperature freeze–thaw splitting test.

## 4. Test Results and Discussion

*4.1. Asphalt Performance Test*

4.1.1. Penetration, Softening Point, and Ductility Test

The asphalt samples are tested for penetration (P) at 25 °C and ductility (D) and softening point (SP) at 10 °C under different freeze–thaw cycles, with the test results shown in Table 4.

**Table 4.** Three indexes of asphalt under different high and low temperature cycles.

| Freeze–Thaw Cycles | No. 90 Asphalt | | | No. 110 Asphalt | | | SBS Modified Asphalt | | |
|---|---|---|---|---|---|---|---|---|---|
| | P/0.1 mm | D/cm | SP/°C | P/0.1 mm | D/cm | SP/°C | P/0.1 mm | D/cm | SP/°C |
| 0 | 89.3 | 92 | 46 | 100.8 | 125 | 43.4 | 73.3 | 79 | 64 |
| 10 | 88.7 | 91.6 | 46.1 | 98.5 | 122.6 | 43.9 | 72.6 | 77.8 | 63.3 |
| 20 | 89.1 | 87.5 | 46.3 | 97.7 | 129.2 | 43.1 | 71.2 | 76.5 | 64.4 |
| 30 | 89 | 88.5 | 45.2 | 98.9 | 123.5 | 44.2 | 71.9 | 75.5 | 64.9 |
| 50 | 87.6 | 88.1 | 46.8 | 97.9 | 121.8 | 44 | 70.7 | 75.9 | 65.7 |
| Maximum rate of change (%) | −1.90 | −4.24 | 1.74 | −2.88 | 3.36 | 1.38 | −3.55 | −4.43 | 2.66 |

As seen from the test results, compared with the test data of the three major indexes of asphalt under no freeze–thaw cycles, the test data of the three major indexes for No. 90 matrix asphalt, No. 110 matrix asphalt, and SBS modified asphalt show few changes, with the rate of change lower than 5%. Under different freeze–thaw cycles, the performance of asphalt basically shows no change before and after the test, mainly because the asphalt undergoes only a physical phase change under the freeze–thaw cycle conditions, with transfer or change delivered in its internal chemical composition.

### 4.1.2. BBR Test

No. 110 asphalt is selected to prepare test samples, and BBR is used to measure the bending stiffness modulus S and the m value of asphalt after freeze–thaw cycles. The test temperature is set to −15 °C. Table 5 shows the test data of stiffness modulus S and m value of No. 110 matrix asphalt before and after 10, 20, 30, and 50 freeze–thaw cycles.

**Table 5.** BBR test results of No. 110 asphalt binder.

| Freeze–Thaw Cycles | 0 | | 10 | | 20 | | 30 | | 50 | |
|---|---|---|---|---|---|---|---|---|---|---|
| Test index | S/MPa | m | S/MPa | m | S/MPa | m | S/MPa | m | S/MPa | m |
| Test results | 187 | 0.361 | 176 | 0.370 | 182 | 0.362 | 186 | 0.359 | 183 | 0.356 |

It can be observed from the test results that after these freeze–thaw cycles, the difference between the stiffness modulus and the m value of the asphalt becomes extremely low. After 10 freeze–thaw cycles, the index difference is the highest, the difference in stiffness modulus is 5.7%, and the difference in m value is 2.6%. Given the possible errors in the test process, it can be concluded that the low-temperature rheological properties of the asphalt have not changed after the freeze–thaw cycles; in other words, the freeze–thaw cycles deliver no effect on the low-temperature properties of the asphalt materials.

### 4.1.3. DSR Test

No. 110 asphalt is selected to prepare test samples, and DSR is used to measure the complex modulus G* and phase angle δ of the asphalt after 10, 20, 30, and 50 freeze–thaw cycles; the rutting factor G*/sinδ is then calculated. In the test, the scanning temperature is set to 46, 52, 58, and 64 °C; the loading frequency is ω = 10 rad/s; and the strain control mode is adopted, with γ = 12%. The test results are shown in Table 6, and the relationship between the rutting factor and temperatures is shown in Figure 8.

**Table 6.** DSR test results of No. 110 matrix asphalt.

| Test Index | Temperature (°C) | Freeze–Thaw Cycles | | | | |
|---|---|---|---|---|---|---|
| | | 0 | 10 | 20 | 30 | 50 |
| G* (kPa) | 46 | 17.08 | 17.39 | 17.25 | 16.02 | 16.58 |
| | 52 | 7.18 | 7.31 | 7.07 | 7.36 | 7.39 |
| | 58 | 3.2 | 3.16 | 3.26 | 3.3 | 3.34 |
| | 64 | 1.47 | 1.34 | 1.51 | 1.51 | 1.5 |
| δ(°) | 46 | 82.22 | 82.92 | 79.75 | 84.69 | 83.25 |
| | 52 | 84.39 | 83.24 | 86.92 | 82.28 | 84.17 |
| | 58 | 86.02 | 86.95 | 88.17 | 83.87 | 87.32 |
| | 64 | 87.31 | 88.25 | 85.13 | 87.4 | 86.53 |
| G*/sinδ (kPa) | 46 | 17.24 | 17.52 | 17.53 | 16.09 | 16.70 |
| | 52 | 7.22 | 7.36 | 7.08 | 7.43 | 7.43 |
| | 58 | 3.2 | 3.16 | 3.27 | 3.31 | 3.34 |
| | 64 | 1.47 | 1.34 | 1.52 | 1.52 | 1.52 |

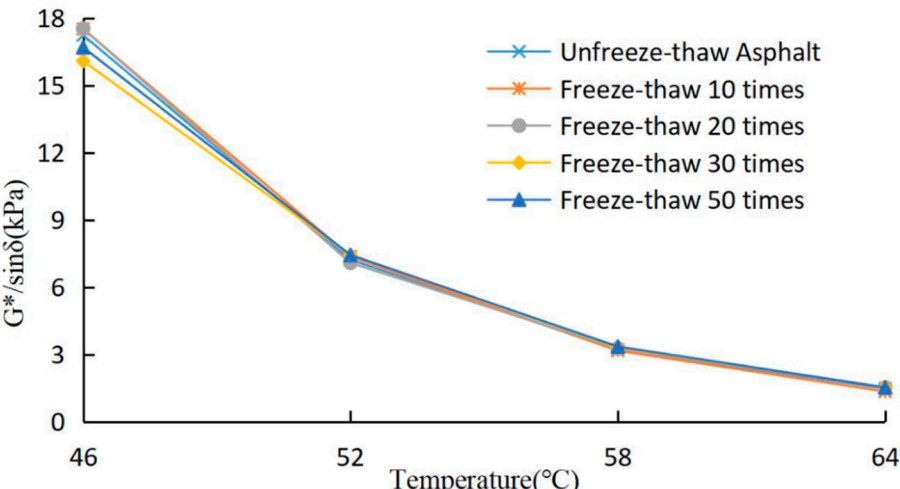

**Figure 8.** Relationship between rutting factor and temperature under high and low temperature cycle of asphalt.

The test results demonstrate that with the increase in temperature, the complex modulus of the asphalt before and after the freeze–thaw cycles would decrease, the phase angle would increase, and the rutting factor would decrease. This is due to the transformation of asphalt from a highly elastic state at low temperature to a viscous fluid state at high temperature. If the viscous component in the viscoelastic properties of asphalt increases, the elastic component will decrease. During this process, the maximum shear stress would go down, whereas the maximum shear strain will go up. After 10, 20, 30, and 50 freeze–thaw cycles, the rutting factor of the asphalt shows very little change compared with the data of the original asphalt. It can be concluded that the rheological properties of the asphalt itself do not change after the freeze–thaw cycles.

### 4.2. Asphalt Mixture Performance Test

#### 4.2.1. TSRST Test

Three kinds of asphalt mixtures, namely matrix asphalt AC-13, SBS modified AC-13, and SBS modified AC-16, are used to prepare the samples, and the temperature stress test for the restrained samples is carried out at a cooling rate of 10 °C/h. Two indicators, i.e., freeze–break temperature and freeze–break stress, are obtained from the test. The test results are illustrated in Figures 9 and 10.

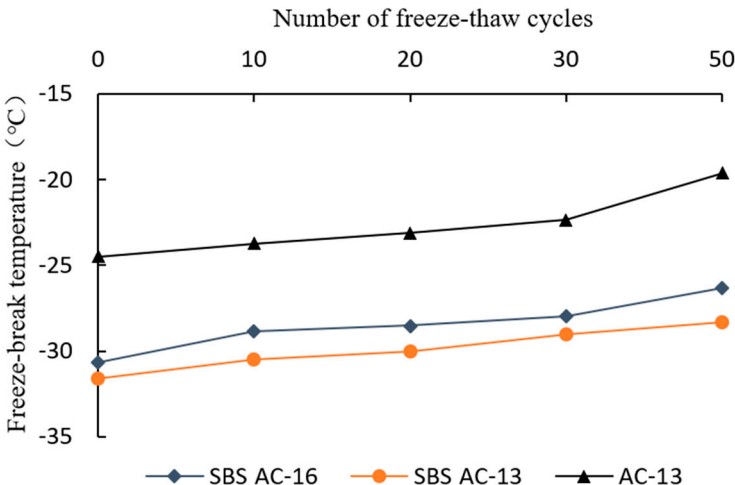

**Figure 9.** Freeze–break temperature test results.

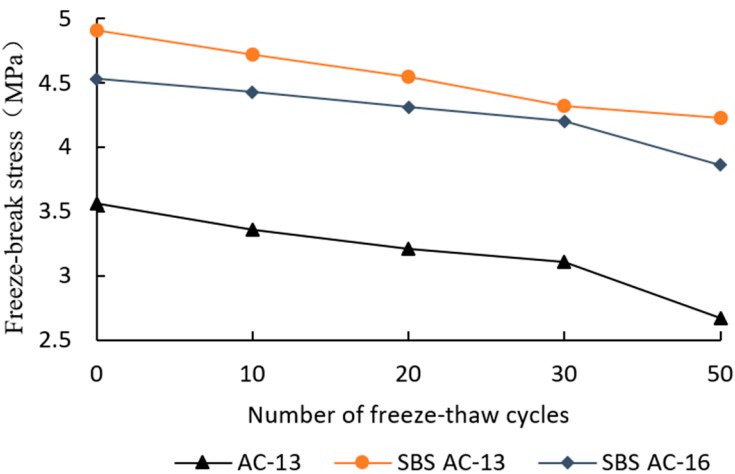

**Figure 10.** Freeze–break stress test results.

As observed, the overall trends of the two indicators of the asphalt mixtures, i.e., freezing–breaking temperature and freezing–breaking stress, are similar under different conditions. The types of mixtures have a certain influence on the test results. The freeze–break temperature of AC-13 asphalt mixture is lower than that of AC-16 asphalt mixture, because the former has smaller particle sizes and larger amounts of asphalt, thus delivering better low-temperature performance than the latter. The types of asphalt also have a certain influence on the test results. The freeze–break temperature of the matrix asphalt is higher than that of the modified asphalt, because the latter has a higher viscosity than the former and, furthermore, the toughness of the latter at a low temperature is better than that of the former.

### 4.2.2. Freeze–Thaw Splitting Test

According to the set plan, the freeze–thaw splitting test is carried out for three kinds of asphalt mixtures, namely matrix asphalt AC-13, SBS modified AC-13, and SBS modified AC-16. Two indicators, i.e., splitting strength and freeze–thaw splitting strength ratio, are obtained from the test calculation. The test results are shown in Table 7.

**Table 7.** Freeze–thaw splitting test results of asphalt mixture.

| Asphalt Mixture | AC-13 | | SBS AC-13 | | SBS AC-16 | |
|---|---|---|---|---|---|---|
| Freeze–Thaw Cycles | Strength (MPa) | Ratio (%) | Strength (MPa) | Ratio (%) | Strength (MPa) | Ratio (%) |
| 0 | 7.39 | 100.00 | 8.65 | 100.00 | 8.73 | 100.00 |
| 10 | 6.14 | 83.09 | 7.88 | 91.10 | 7.85 | 89.92 |
| 20 | 5.66 | 76.59 | 7.46 | 86.24 | 7.54 | 86.37 |
| 30 | 5.42 | 73.34 | 7.13 | 82.43 | 7.12 | 81.56 |
| 50 | 5.16 | 69.82 | 6.89 | 79.65 | 6.83 | 78.24 |

It can be seen from the test results that the splitting strength of asphalt mixtures and the ratio of freeze–thaw splitting test strength are significantly decreased under the condition of frequent freeze–thaw cycles; with the increase in the number of freeze–thaw cycles, the splitting strength and the ratio of freeze–thaw splitting test strength would gradually go down. These results demonstrated that frequent freeze–thaw cycles would deliver a significant effect on the splitting tensile strength and water stability of asphalt mixtures. As the freeze–thaw splitting test is carried out in a water-containing state, when the temperature drops below 0 °C, the water in the asphalt mixtures will begin to freeze and expand, causing cracks, or the original cracks will keep expanding. When the temperature rises above 0 °C, the ice within the sample would be re-thawed, and the

water will continuously infiltrate into the new cracks. Then, the process of freezing and expanding will begin again in the next freeze–thaw cycle, which can deliver a significant impact on the low-temperature crack resistance and water stability of the asphalt mixtures.

## 5. Conclusions

This study explores the freeze–thaw cycles' characteristics and the cracking problems of asphalt pavements in high-cold and high-altitude areas. Laboratory tests are made to evaluate the pavements' performance of asphalt binder as well as the performance of low-temperature crack resistance of asphalt mixtures under frequent freeze–thaw cycles. The main conclusions are as follows.

(1) There are basically more than 120 freeze–thaw cycles per year in the high-cold and high-altitude areas, or even more than 200 cycles. With the increase in altitudes, the number of freeze–thaw cycles shows an up-ticking trend. Frequent freeze–thaw cycles have a significant impact on asphalt pavements, the cracks in which have become a typical problem.

(2) The performances of the three different types of asphalt binders used in the test basically show no change after 50 freeze–thaw cycles, mainly because under such cycles, asphalt binders will only change its physical phases, but its internal chemical composition does not transfer or change, so neither the properties of the asphalt.

(3) The asphalt types have a significant effect on the low-temperature performance of asphalt mixtures. The modified asphalt shows a higher viscosity than the matrix asphalt, with better toughness than that of the matrix asphalt at low temperature. This result demonstrates that the freeze–break temperature of the matrix asphalt is higher than that of the modified asphalt.

(4) Frequent freeze–thaw cycles significantly influence the low-temperature splitting tensile strength and water stability of asphalt mixtures. With increased freeze–thaw cycles, the splitting strength and freeze–thaw splitting tensile strength ratio will gradually decrease to a significant level.

**Author Contributions:** H.C.: conceptualization, methodology, validation, investigation, writing—original draft. T.C.: conceptualization, writing—review and editing, supervision, project administration, resources, funding acquisition. H.Z.: methodology, writing—review and editing. H.R.: writing—review and editing, supervision. All authors have read and agreed to the published version of the manuscript.

**Funding:** This research was funded by National Natural Science Foundation of China (Grant No. 52178419), Transportation Industry Key Science and Technology Projects of China (Grant No. 2020-MS1-059) and Youth Science and Technology Innovation Fund project of FHCC (Grant No. YGY2019QC-02).

**Institutional Review Board Statement:** Not applicable.

**Informed Consent Statement:** Not applicable.

**Data Availability Statement:** Data will be made available on reasonable request.

**Acknowledgments:** Thanks to Zhendong Qian, who proposed the idea of the performance test in the current study.

**Conflicts of Interest:** The authors declare no conflict of interest.

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
