# Peer review of "Influence of Frequent Freeze–Thaw Cycles on Performance of Asphalt Pavement in High-Cold and High-Altitude Areas"

_coatings, doi:10.3390/coatings12060752_

Round 1
Reviewer 1 Report
The article demonstrates valuable and interesting research results both experimental and theoretical. In the available literature on the subject, there is little available work on the operational properties of asphalt surfaces at high altitudes with large temperature amplitudes. In this aspect, the work is an interesting item, although some some of the conclusions can be expected a priori.
In the toretic part, I lacked information about the places of initiation of cracks and the mechanisms of propagation. The authors focused on the freezing and thawing process. The elements related to the operational load of the asvalt surface as well as the quality of preparation of the surface base were omitted. This is a kind of simplification. Fig. 3. After what period of operation the cracks appeared (after what number of cycles of freezing - thawing). What was the operational load and what is the base of the surface? Without these data, evaluation is not possible. What were the criteria for selecting asphalt mixes for run-offs - it is not clear to me. Complete the data with information about the device on which the BBR and DSR test was made.Due to the fact that the research was carried out on the experimental TSRST system, the authors should include a diagram of such a device. Are the authors able to indicate the place of initiation to crack?
Reviewer 2 Report
Summary:
This study conducted experiments on the asphalt and asphalt mixtures to investigate the effect of frequency of the freeze-thaw cycles as well as low temperature on their performance such as crack resistance. They reported no significant effect of the frequency of the cycles whereas high impact of the low temperature on the asphalt properties depending on its material. The modified asphalt showed better resistance toward low temperature in compared to the mixture. By increasing the freeze-thaw cycles, the splitting strength and the tensile strength ration will decrease. The finding of this research can be useful for pavement design for the regions with cold climates or high variations in the temperature during the year.
Major comments:
- Abstract needs to be revised to emphasize on the finding of the study rather than introduction.
- Regarding the technical index values provided in the Table 1, was these values measured? If not the reference needs to be provided.
- Reference needs to be added for the data provided in Table 2.
- Line 193: “optimum oil-to-stone ratio of different asphalt mixtures is determined by the Marshall test method”. The detailed calculation and the approach needs to be added to the manuscript.
- Line 199: The Freeze-thaw cycles method was conducted in a environmental test chamber. Was the chamber designed in house? Or was purchase from the manufacturer? What are the properties of the chamber? What is the humidity inside the chamber? Was it measured and controlled? The humidity can have a significant effect on the asphalt cracking as it was reported by previous research articles.
- Line 199: In the experimental design for the “Freeze-thaw cycles method”, please add explanation the reason of choosing the temperature set points? Based on the data provided in the Figure 1, the temperature span was different than what is was chosen. Is 50 degree realistic temperature based on the available data?
- Line 212: The instrumental information for the conducted measurements needs to be added. What instruments were used for obtaining each indices? The same for other sections.
- Line 295 and 308: Is there any specific reason that the BPR, DSR test were not conducted on other asphalts and No. 110 was chosen?
- The number of freeze-thaw cycles were reported in the manuscript in multiple sections; however, this information was not utilized in the experimental design. Please add justification for choosing the number of cycles.
Minor comments:
- The quality of the images in Figures are very low.
Reviewer 3 Report
The research explored the temperature changes and freeze-thaw cycles in certain high-altitude areas. The topic is very interesting, and the paper is well written. To accept the paper for publication the reviewer suggest to improve it on some minor aspects as below reported.
-Table 1 It is suggested to insert a column for indicating the units,
-What does technology require mean reported in table 2?
-Please, provide more information regarding the asphalt performance tests. An example is the following study that would be useful if cited. Coatings 2021, 11(7), 751; https://doi.org/10.3390/coatings11070751
-It is requested to deepen the results obtained and if possible to create a diagram at the end of the discussions that correlates all the performances analyzed in this study
